# The prevalence and demographic features of congenital cytomegalovirus infection in an urban area of East Asia: A population-based study

**Tzong-Hann Yang**[1,2,3], **Hung-Meng Huang**[1,4], **Wei-Chung Hsu**[5], **Po-Nien Tsao**[6], **Tien-Chen Liu**[5], **Chuan-Jen Hsu**[5], **Li-Min Huang**[6], **Chuan-Song Wu**[1,7], **Shih-Ming Weng**[2,8], **Chun-Yi Lu**[6]*, **Chen-Chi Wu**[5,9]*

**1** Department of Otorhinolaryngology, Taipei City Hospital, Taipei, Taiwan, **2** Department of Speech Language Pathology and Audiology, National Taipei University of Nursing Health Sciences, Taipei, Taiwan, **3** General Education Center, University of Taipei, Taipei, Taiwan, **4** Department of Otolaryngology, School of Medicine, College of Medicine, Taipei Medical University, Taipei, Taiwan, **5** Department of Otolaryngology, National Taiwan University Hospital, Taipei, Taiwan, **6** Department of Pediatrics, National Taiwan University Hospital, Taipei, Taiwan, **7** College of Science and Engineering, Fu Jen University, New Taipei City, Taiwan, **8** Department of Pediatrics, Wan Fang Hospital, Taipei Medical University, Taipei, Taiwan, **9** Department of Medical Research, National Taiwan University Biomedical Park Hospital, Hsinchu, Taiwan

* cylu@ntu.edu.tw (C-YL); chenchiwu@ntuh.gov.tw (C-CW)

## Abstract

Congenital cytomegalovirus (cCMV) infection is the leading environmental cause of child-hood hearing impairment. However, its significance remains largely undocumented in many regions of the world. The purpose of this study was to investigate the prevalence and clinical features of cCMV infection in East Asia. Neonates born at a municipal hospital in Taipei were prospectively recruited and underwent concurrent hearing and CMV screenings. Those who failed the hearing screening or screened positive for CMV were subjected to a focused audiological and/or virological surveillance. The characteristics of the newborns and their mothers were compared between the CMV-positive and CMV-negative groups. Of the 1,532 newborns who underwent concurrent hearing and CMV screenings, seven (0.46%) were positive for cCMV infection. All seven CMV-positive newborns were asymptomatic at birth, and none of them developed hearing or other symptoms during a follow-up period of 14.4±6.3 months. The mothers of the CMV-positive newborns demonstrated higher gravidity (2.4 ± 1.4 vs. 2.1 ± 1.2) and parity (2.0 ± 1.2 vs. 1.6 ± 0.7) than those in the CMV-negative group; however, the difference did not reach statistical significance. The prevalence of cCMV infection in Taipei newborns was 0.46%, which is slightly lower than that of other populations and that of a previous report in the Taiwanese population. The relatively low prevalence in this study might be attributed to the improved public health system and decreased fertility rate in Taiwan.

**Data Availability Statement:** All relevant data are within the paper and its Supporting information files.

**Funding:** The study was supported by research grants from the National Taiwan University Hospital (107-S3913), Taipei City Government (10501- 62-030), and the Ministry of Science and Technology, Taiwan (107-2622-B-002-008-CC2.

**Competing interests:** The authors have declared that no competing interests exist.

## Introduction

Sensorineural hearing loss (SNHL) is a common clinical entity in newborns [1,2] and children [3,4]. Pediatric SNHL is an etiologically heterogeneous condition caused by a plethora of genetic [5–7] and environmental factors [8–12]. Recent advances in molecular genetics have revolutionized the assessment armamentarium of pediatric SNHL, enabling us to ascertain the etiology in 40–60% of the children with SNHL [13,14]. However, the clinical significance and contribution of the environmental factors that might lead to pediatric SNHL in these children largely remains unclear [8,11,12,15,16].

Among these environmental factors, congenital cytomegalovirus (cCMV) infection is the leading cause of pediatric SNHL [17] and neurodevelopmental disability [18] in developed countries. The importance of identifying cCMV infection as the etiology of SNHL in newborns has become clinically relevant with the availability of oral antiviral agents that may prevent the progression of cCMV-related SNHL [19]. Furthermore, children with cCMV infection are at risk for progressive SNHL that may not be present until several years of age, at a time when the golden period for hearing-loss rehabilitation has passed [20–22].

The prevalence and clinical characteristics of cCMV-infected newborns have been reported in several Western series [17,23]. However, there is still a paucity of such reports in East Asia, a populated region with rapid economic development. In the present study, we aimed to investigate the prevalence of cCMV infection in newborns from an urban region of East Asia and the clinical characteristics of cCMV-positive children and their mothers.

## Materials and methods

### Subject recruitment

From May 2016 to Dec 2018, we prospectively enrolled neonates born at the Taipei City Hospital Fuyou Branch. All newborns underwent hearing screening using automated auditory brainstem response (AABR) testing [24], and saliva swabs were obtained simultaneously for cCMV screening.

For the newborns who screened positive for cCMV, the following variables were recorded: sex, mode of delivery, age at last follow-up, birth weight, maternal age, gestational age, mother's gravidity and parity, and presence of symptoms/signs at birth, such as newborn hearing screening (NHS) failure or neonatal jaundice, and admission status were recorded.

All infants were of Han Taiwanese ethnicity. Written informed consent was obtained from the parents of all infants. This study was approved by the Research Ethics Committees of Taipei City Hospital and the National Taiwan University Hospital.

### CMV screening

CMV screening was performed using a quantitative real-time polymerase chain reaction (PCR) assay with fluorescence resonance energy transfer (FRET) hybridization probes to detect the glycoprotein B of CMV [25]. The lower limit of detection, estimated using a CMV construct, was 10 copies/ml. All positive results were replicated in a second test, and samples that tested positive in both were considered true positives. Positive CMV PCR results were then confirmed by isolating CMV from urine or saliva.

### Audiologic and clinical assessments in CMV-positive infants

Infants who tested positive for CMV at birth were subjected to a focused audiologic surveillance, including repeated AABR testing at 1 month, followed by comprehensive audiologic assessments at 3 months, 6 months, and 1 year [25].

These infants also underwent additional clinical evaluations, including complete blood counts, blood biochemistry, brain transfontanellar ultrasonography, abdominal ultrasonography, neurologic assessment, and visual assessment. Virological tests, including the determination of CMV viral loads in the blood using quantitative real-time PCR and the detection of CMV from a culture of bodily fluids (either urine or a throat swab), were performed during every medical examination to monitor viral clearance [25].

### Data analyses

The results of the CMV screening were compared to the NHS results. The characteristics of the newborns and their mothers were analyzed according to their sex, mode of delivery, gestational age, birth weight, maternal age at pregnancy, gravidity, and parity. The proportions between the groups were compared using Fisher's exact test. All analyses were conducted using SAS software, version 9.3 (SAS Institute, Inc., Cary, NC).

## Results

During the study period, 3,273 neonates were born at the Taipei City Hospital Fuyou Branch (Fig 1). Of these, the parents of 1,532 neonates agreed to undergo a newborn CMV screening for their children. The CMV screening was positive in seven newborns (0.46%), including one girl and six boys (Table 1). CMV infection was confirmed in all seven newborns by isolation of CMV from saliva and/or urine. All seven CMV-positive newborns passed the NHS. Of the other 1,525 infants who were negative for CMV, 25 (1.6%) failed the initial NHS and three (0.2%) were subsequently confirmed to have unilateral or bilateral SNHL. Among the 1,741 infants who did not undergo CMV screening, 64 (3.7%) failed the initial NHS and 16 (0.9%) were subsequently confirmed to have SNHL. In total, the prevalence of neonatal cCMV infection was 0.46%, with 0.74% for males and 0.14% for females. There was no statistically significant difference in the prevalence between the sexes.

The characteristics of the seven CMV-positive newborns are presented in Table 2. The newborns of cases 2 and 3 were given birth by a cesarean section (2/7, 28.6%), while the other five newborns were given birth through a normal spontaneous delivery (5/7, 71.4%).

All seven newborns with cCMV infection were asymptomatic at birth, and none of them were admitted to a neonatal intensive care unit or developed hyperbilirubinemia that required a specific treatment, including phototherapy or plasmapheresis (Table 2). The newborns of cases 2 and 4 were lost to follow-up for unknown reasons. The other five newborns were followed up at the National Taiwan University Hospital for a mean duration of 14.4±6.3 months. None of these five newborns developed hearing or other symptoms during the follow-up period.

Table 3 presents a comparison of the maternal/neonatal characteristics between newborns with and without cCMV infection. The mothers of the CMV-positive neonates had a slightly higher gravidity (2.4 ± 1.4 vs. 2.1 ± 1.2, respectively) and parity (2.0 ± 1.2 vs. 1.6 ± 0.7, respectively) than those whose newborn babies were screened negative for CMV; yet, the difference did not reach statistical significance.

## Discussion

In this study, we demonstrated that the prevalence of cCMV infection is approximately 0.46% in Taipei, a typical populous city in East Asia. The rate is slightly lower than that in other populations. In developed countries, the prevalence of cCMV infection ranged from 0.58% to 0.70% [17,26,27]. In developing countries, such as Mexico, Nigeria, and Gambia, the prevalence of cCMV infection was higher than that in our study, ranging from 0.9% to 5.4% [28–32]. Several

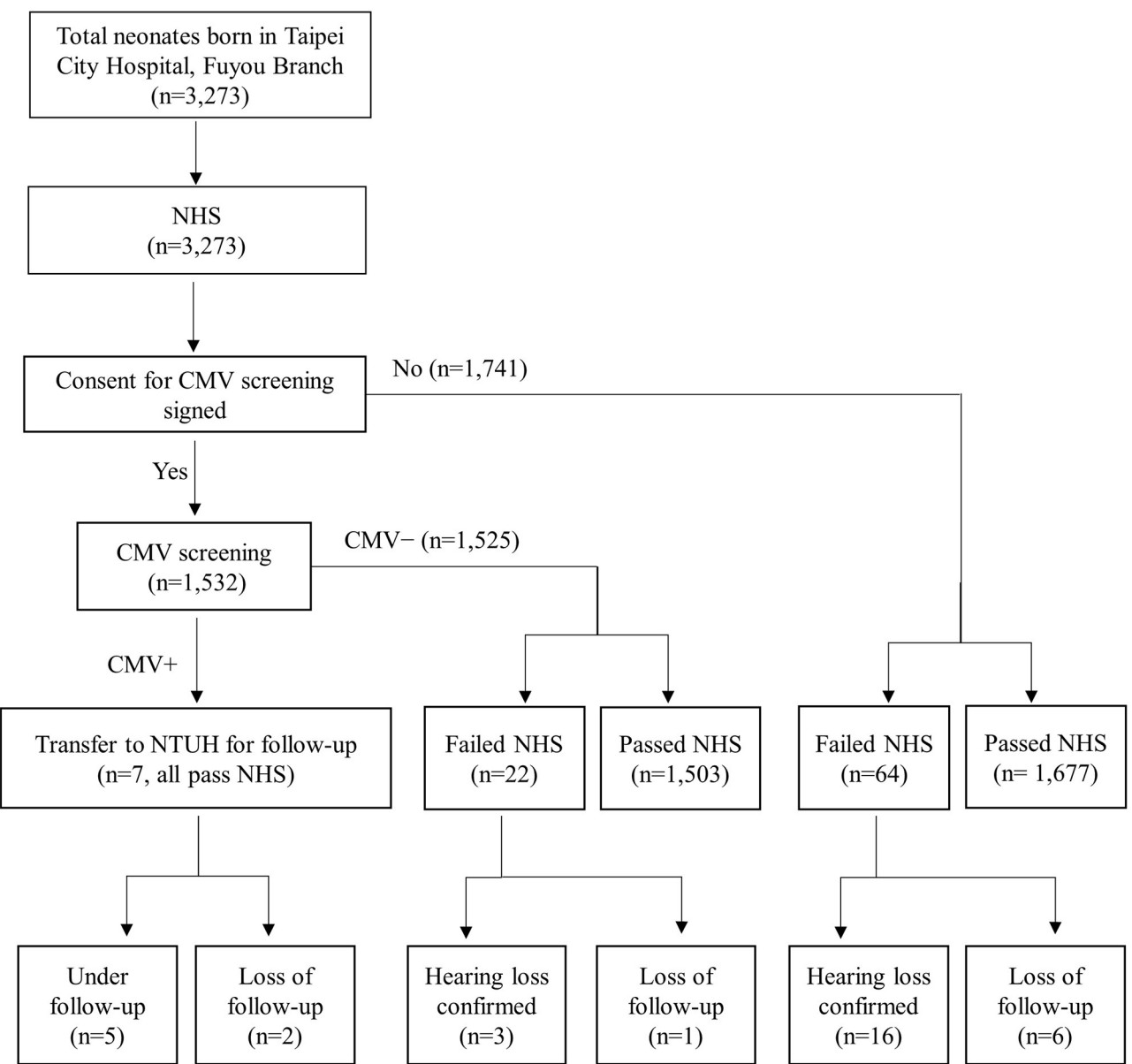

**Fig 1. Flow diagram of the recruitment and the outcome of the newborn hearing screening (NHS) and cCMV screening in 3,273 newborns in Taipei, Taiwan.** NTUH denotes National Taiwan University Hospital.

**Table 1. Prevalence of cCMV infection in the 1,532 newborns.**

|  | cCMV+ | cCMV- | Total | Prevalence | P value |
|---|---|---|---|---|---|
| Total number | 7 | 1,525 | 1,532 | 0.46% |  |
| Sex |  |  |  |  | 0.129 |
| Male | 6 | 805 | 811 | 0.74% |  |
| Female | 1 | 721 | 722 | 0.14% |  |
| Mean age at last follow-up, m | 14.4±7.1 |  |  |  |  |

**Table 2. Characteristics of the 7 newborns with cCMV infection.**

| Case | Sex | Mode of delivery | Age at last follow-up, m | Age of mother | Gestational age (weeks) | Gravidity | Parity | Birth weight in kg (%) | Head girth at birth in cm (%) | Hearing level | Other symptoms |
|---|---|---|---|---|---|---|---|---|---|---|---|
| 1 | F | NSD | 19 | 19 | 39 | 1 | 1 | 3.50 (50–85) | 36 (50–85) | normal | none |
| 2 | M | CS | lost to follow-up | 39 | 38 | 2 | 2 | 3.14 (15–85) | 35 (50–85) | NA | NA |
| 3 | M | CS | 18 | 41 | 38 | 5 | 4 | 3.70 (50–85) | 36 (50–85) | normal | none |
| 4 | M | NSD | lost to follow-up | 37 | 39 | 2 | 2 | 3.90 (50–85) | 35.5 (50–85) | NA | NA |
| 5 | M | NSD | 18 | 26 | 34 | 3 | 3 | 3.36 (50–85) | 34 (15–50) | normal | none |
| 6 | M | NSD | 15 | 51 | 39 | 3 | 1 | 3.58 (50–85) | 36 (50–85) | normal | none |
| 7 | M | NSD | 2 | 27 | 37 | 1 | 1 | 2.78 (3–15) | 33 (15–50) | normal | none |

CS, cesarean section; NA, not available; NSD, normal spontaneous delivery.

factors have been proposed to account for the differences in the prevalence of cCMV infections in newborns between the developed and developing countries, including the testing methods [26], sampling methodology [26], selection bias [26], and seroprevalence rate in the general population [31,33–35], and ethnicity [26,36,37].

Noticeably, in a prior study in 1996, the prevalence of cCMV infection in Taiwanese newborns was found to be 1.8% [38]. The significantly lower prevalence in the present study might reflect the improved public health conditions, including better prenatal care in Taiwan over the past two decades. Interestingly, a recent study in China also reported a low cCMV infection prevalence (0.7%, n = 10.933) [33] as compared to that in three other large-scale studies in Portuguese (1.05%, n = 3,600) [34], Brazilian (1.1%, n = 8,047) [35], and Turkish (1.9%, n = 944) [31] populations, although the seroprevalence of CMV is commensurately high among the four populations. The authors attributed the lower cCMV infection in the Chinese population to the lower exposure of pregnant women to young children than in the other populations as a result of China's unique one-child policy [33]. In our study, we also observed a higher parity number in the mothers with CMV-positive newborns (2.0 ± 1.2) than in those with CMV-negative newborns (1.6 ± 0.7). It is thus likely that the low fertility rate, that is, the number of children a woman is expected to have during her childbearing years, might also contribute to the low cCMV infection rate in the Taiwanese newborns in the present study. Since 2008, one of the lowest fertility rates (0.895–1.27) among any territory in the world has been recorded in the Taiwanese population (https://eng.stat.gov.tw/public/data/dgbas03/bs2/yearbook_eng/y005.pdf).

**Table 3. Comparison of maternal/neonatal characteristics between subjects with and those without cCMV infection.**

| Characteristics | CMV+(n = 7) | CMV-(n = 1,525) | P value |
|---|---|---|---|
| Gender (male) | 6 (85.7%) | 804 (52.7%) | .083 |
| Mode of delivery (NSD) | 5 (71.4%) | 1041 (68.3%) | .608 |
| Birth weight (kg) | 3.3 ± 0.3 | 3.1 ± 0.4 | .088 |
| Age of mother (y) | 34.4 ± 10.9 | 33.2 ± 4.7 | .768 |
| Gestational age (weeks) | 38.3 ± 0.8 | 38.5 ± 1.2 | .643 |
| Gravidity | 2.4 ± 1.4 | 2.1 ± 1.2 | .509 |
| Parity | 2.0 ± 1.2 | 1.6 ± 0.7 | .154 |
| Failed NHS | 3 (0.5%) | 0 (0%) | .986 |

NHS: Newborn hearing screen; NSD: Normal spontaneous delivery.

The screening tool for cCMV infection is also crucial when the prevalence of cCMV infection is determined and compared across populations. It has been demonstrated that real-time PCR assays on saliva samples collected from live-born newborns performed within 2 days from birth, as performed in the present study, enables high sensitivity and specificity for identifying a cCMV infection in newborns [39,40]. In contrast, real-time PCR assay on dry blood-spot samples had a low sensitivity, restraining its value as a screening test for cCMV infection [41]. Our earlier work also showed a lower rate of cCMV infection (0.17%, 3/1,716) in a similar population when PCR on dried blood spot (DBS) specimens was used as a screening method [25].

Of note, although all seven newborns who screened positive for cCMV infection passed the hearing screening at birth and none of them developed a hearing impairment during the follow-up period, a regular and continual check-up on the hearing level and viral status is necessary in these infants, as it has been documented that a certain proportion of infants with asymptomatic cCMV infection might develop a hearing impairment or other sequelae later on, even as late as at the adolescent age [22].

The strength of this study is that all subjects were recruited from a community-based hospital, which offered an unbiased estimation of the representative prevalence of cCMV infection among newborns in Taipei. All the CMV PCR and confirmatory CMV isolation tests were fully supported by research grants and were hence free to parents. This further prevented selection biases brought about by socioeconomic status. The composition of newborns in the present study is different from that recruited from medical centers [26], where the recruitment of subjects could be biased to a study population with a higher risk of disease propensity.

However, some limitations of this study deserve discussion. First, the current study included a sample size of 1,532 newborns and an average follow-up period of 14.4 months. A larger series with a longer follow-up period will possibly disclose the prevalence and clinical outcomes of cCMV infection with greater precision. Second, the notion that low cCMV infection prevalence is associated with low fertility rates needs to be tested in other populations with low fertility rates. Further investigations in other East Asian populations with a low fertility rate, such as the Korean and Japanese populations, are highly expected.

## Conclusion

The prevalence of cCMV infection in newborns of Taipei, Taiwan is 0.46%, which is slightly lower than that in other populations and that previously reported in the Taiwanese population. The relatively low prevalence in this study might be attributed to the improved public health and prenatal care system and the decreased fertility rate in Taiwan.

## Supporting information

**S1 File.**
(XLSX)

## Author Contributions

**Conceptualization:** Chun-Yi Lu, Chen-Chi Wu.

**Data curation:** Tzong-Hann Yang.

**Formal analysis:** Tzong-Hann Yang.

**Funding acquisition:** Tzong-Hann Yang, Hung-Meng Huang, Chuan-Song Wu, Chen-Chi Wu.

**Investigation:** Tzong-Hann Yang, Chuan-Song Wu, Chun-Yi Lu.

**Project administration:** Tzong-Hann Yang.

**Resources:** Tien-Chen Liu.

**Software:** Wei-Chung Hsu, Li-Min Huang, Shih-Ming Weng.

**Supervision:** Hung-Meng Huang, Po-Nien Tsao, Tien-Chen Liu, Chuan-Jen Hsu, Chuan-Song Wu, Chun-Yi Lu.

**Validation:** Chuan-Jen Hsu.

**Visualization:** Hung-Meng Huang, Wei-Chung Hsu, Po-Nien Tsao, Chuan-Jen Hsu, Li-Min Huang, Chuan-Song Wu, Shih-Ming Weng.

**Writing – original draft:** Tzong-Hann Yang.

**Writing – review & editing:** Chun-Yi Lu, Chen-Chi Wu.

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
