## [Decision Letter · Decision Letter 0]

30 Oct 2020

PONE-D-20-23918

The Prevalence and Hearing Outcome of Congenital CMV Infection in an Urban Area of East Asia: Population-based Study

PLOS ONE

Dear Dr. Wu,

Thank you for submitting your manuscript to PLOS ONE. After careful consideration, we feel that it has merit but does not fully meet PLOS ONE’s publication criteria as it currently stands. Therefore, we invite you to submit a revised version of the manuscript that addresses the points raised during the review process.

We look forward to receiving your revised manuscript.

Kind regards,

Marta Giovanetti, Ph.D.

Academic Editor

PLOS ONE

Journal Requirements:

"The study was supported by research grants from the National Taiwan University Hospital (107-S3913), Taipei City Government (10501-62-030), and the Ministry of Science and Technology, Taiwan (107-2622-B-002-008-CC2)."

 "No."

[copy in statement].

Reviewers' comments:

Reviewer's Responses to Questions

**Comments to the Author**

1. Is the manuscript technically sound, and do the data support the conclusions?

Reviewer #1: No

Reviewer #2: Yes

2. Has the statistical analysis been performed appropriately and rigorously? 

Reviewer #1: No

Reviewer #2: Yes

3. Have the authors made all data underlying the findings in their manuscript fully available?

Reviewer #1: No

Reviewer #2: Yes

4. Is the manuscript presented in an intelligible fashion and written in standard English?

Reviewer #1: No

Reviewer #2: Yes

5. Review Comments to the Author

Reviewer #1: The article presents a unique study focused on a locality (Taipei, Taiwan), aiming at a general and comprehensive background of East Asia. No details were given on the choice of the hospital to conduct the research. A more comprehensive study would be needed for the effect of cCMV infection in the population assessment.

The figure needs to improve resolution.

Reviewer #2: "The Prevalence and Hearing Outcome of Congenital CMV Infection in an Urban Area of East Asia: Population-based Study" by Yang et al. is well-written and provides a relatively comprehensive view about Congenital cytomegalovirus prevalence in Taipei, Taiwan. Although the study has limitations already recognized by the authors, this manuscript provide advances in our understanding about the relevance the early identification of cCMV in neonates increasing the quality of life of patients and better follow-up of the public service to these patients telling a nice story. I think it would be suitable for publication in PLoS One after careful revision of English language.

6. PLOS authors have the option to publish the peer review history of their article (what does this mean?). If published, this will include your full peer review and any attached files.

Reviewer #1: No

Reviewer #2: No

---

## [Author Response · Author response to Decision Letter 0]

18 Jan 2021

PONE-D-20-23918

The Prevalence and Hearing Outcome of Congenital CMV Infection in an Urban Area of East Asia: Population-based Study

PLOS ONE

Dear Dr. Wu,

Thank you for submitting your manuscript to PLOS ONE. After careful consideration, we feel that it has merit but does not fully meet PLOS ONE’s publication criteria as it currently stands. Therefore, we invite you to submit a revised version of the manuscript that addresses the points raised during the review process.

We look forward to receiving your revised manuscript.

Kind regards,

Marta Giovanetti, Ph.D.

Academic Editor

PLOS ONE

Journal Requirements:

"The study was supported by research grants from the National Taiwan University Hospital (107-S3913), Taipei City Government (10501-62-030), and the Ministry of Science and Technology, Taiwan (107-2622-B-002-008-CC2)."

 "No."

[copy in statement].

Reviewers' comments:

Reviewer's Responses to Questions

Comments to the Author

1. Is the manuscript technically sound, and do the data support the conclusions?

Reviewer #1: No

Reviewer #2: Yes

2. Has the statistical analysis been performed appropriately and rigorously?

Reviewer #1: No

Reviewer #2: Yes

3. Have the authors made all data underlying the findings in their manuscript fully available?

Reviewer #1: No

Reviewer #2: Yes

4. Is the manuscript presented in an intelligible fashion and written in standard English?

Reviewer #1: No

Reviewer #2: Yes

5. Review Comments to the Author

Reviewer #1: The article presents a unique study focused on a locality (Taipei, Taiwan), aiming at a general and comprehensive background of East Asia. No details were given on the choice of the hospital to conduct the research. A more comprehensive study would be needed for the effect of cCMV infection in the population assessment.

The figure needs to improve resolution.

Reviewer #2: "The Prevalence and Hearing Outcome of Congenital CMV Infection in an Urban Area of East Asia: Population-based Study" by Yang et al. is well-written and provides a relatively comprehensive view about Congenital cytomegalovirus prevalence in Taipei, Taiwan. Although the study has limitations already recognized by the authors, this manuscript provide advances in our understanding about the relevance the early identification of cCMV in neonates increasing the quality of life of patients and better follow-up of the public service to these patients telling a nice story. I think it would be suitable for publication in PLoS One after careful revision of English language.

6. PLOS authors have the option to publish the peer review history of their article (what does this mean?). If published, this will include your full peer review and any attached files.

Do you want your identity to be public for this peer review? For information about this choice, including consent withdrawal, please see our Privacy Policy.

Reviewer #1: No

Reviewer #2: No

---

## [Decision Letter · Decision Letter 1]

8 Mar 2021

The prevalence and demographic features of congenital cytomegalovirus infection in an urban area of East Asia: a population-based study

PONE-D-20-23918R1

Dear Dr. Wu,

We’re pleased to inform you that your manuscript has been judged scientifically suitable for publication and will be formally accepted for publication once it meets all outstanding technical requirements.

Kind regards,

Kazumichi Fujioka

Academic Editor

PLOS ONE

Additional Editor Comments (optional):

Reviewers' comments:

Reviewer's Responses to Questions

**Comments to the Author**

1. If the authors have adequately addressed your comments raised in a previous round of review and you feel that this manuscript is now acceptable for publication, you may indicate that here to bypass the “Comments to the Author” section, enter your conflict of interest statement in the “Confidential to Editor” section, and submit your "Accept" recommendation.

Reviewer #1: All comments have been addressed

Reviewer #2: All comments have been addressed

2. Is the manuscript technically sound, and do the data support the conclusions?

Reviewer #1: Yes

Reviewer #2: Yes

3. Has the statistical analysis been performed appropriately and rigorously? 

Reviewer #1: N/A

Reviewer #2: Yes

4. Have the authors made all data underlying the findings in their manuscript fully available?

Reviewer #1: Yes

Reviewer #2: Yes

5. Is the manuscript presented in an intelligible fashion and written in standard English?

Reviewer #1: Yes

Reviewer #2: Yes

6. Review Comments to the Author

Reviewer #1: (No Response)

Reviewer #2: "The Prevalence and Hearing Outcome of Congenital CMV Infection in an Urban Area of East Asia: Population-based Study" by Yang et al. I had suggested the acceptance of the article by PLoS One after a rigorous review of English, I maintain my decision.

7. PLOS authors have the option to publish the peer review history of their article (what does this mean?). If published, this will include your full peer review and any attached files.

Reviewer #1: No

Reviewer #2: No

---

## [Editor Report · Acceptance letter]

16 Mar 2021

PONE-D-20-23918R1 

The prevalence and demographic features of congenital cytomegalovirus infection in an urban area of East Asia: a population-based study 

Dear Dr. Wu:

I'm pleased to inform you that your manuscript has been deemed suitable for publication in PLOS ONE. Congratulations! Your manuscript is now with our production department. 

Kind regards, 

on behalf of

Dr. Kazumichi Fujioka 

Academic Editor

PLOS ONE